# Investigation of dual antiplatelet therapy after coronary stenting in patients with chronic kidney disease

Chih-Chin Kao[1,2,3], Mai-Szu Wu[1,3,4], Ming-Tsang Chuang[5], Yi-Cheng Lin[6,7], Chun-Yao Huang[8,9,10], Wei-Chiao Chang[4,11,12], Chih-Wei Chen[9,10]* , Tzu-Hao Chang[13,14]

1 Division of Nephrology, Department of Internal Medicine, School of Medicine, College of Medicine, Taipei Medical University, Taipei, Taiwan, 2 Division of Nephrology, Department of Internal Medicine, Taipei Medical University Hospital, Taipei, Taiwan, 3 TMU Research Center of Urology and Kidney (TMU-RCUK), Taipei Medical University, Taipei, Taiwan, 4 Division of Nephrology, Department of Internal Medicine, Shuang-Ho Hospital, Taipei Medical University, New Taipei City, Taiwan, 5 Clinical Data Center, Office of Data Science, Taipei Medical University, Taipei, Taiwan, 6 Department of Pharmacy, Taipei Medical University Hospital, Taipei, Taiwan, 7 School of Pharmacy, College of Pharmacy, Taipei Medical University, Taipei, Taiwan, 8 Division of Cardiology, Department of Internal Medicine, School of Medicine, College of Medicine, Taipei Medical University, Taipei, Taiwan, 9 Division of Cardiology, Department of Internal Medicine and Cardiovascular Research Center, Taipei Medical University Hospital, Taipei, Taiwan, 10 Taipei Heart institute, Taipei Medical University, Taipei, Taiwan, 11 Department of Clinical Pharmacy, School of Pharmacy, Taipei Medical University, Taipei, Taiwan, 12 Master Program for Clinical Pharmacogenomics and Pharmacoproteomics, School of Pharmacy, Taipei Medical University, Taipei, Taiwan, 13 Graduate Institute of Biomedical Informatics, College of Medical Science and Technology, Taipei Medical University, Taipei, Taiwan, 14 Clinical Big Data Research Center, Taipei Medical University Hospital, Taipei, Taiwan

☯ These authors contributed equally to this work.

* ians.yahoo@gmail.com

**Data Availability Statement:** All relevant data are within the manuscript and its Supporting information files.

## Abstract

### Background

Dual antiplatelet therapy (DAPT) is currently the standard treatment for the prevention of ischemic events after stent implantation. However, the optimal DAPT duration remains elusive for patients with chronic kidney disease (CKD). Therefore, we aimed to compare the effectiveness and safety between long-term and short-term DAPT after coronary stenting in patients with CKD.

### Methods

This retrospective cohort study analyze data from the Taipei Medical University (TMU) Institutional and Clinical Database, which include anonymized electronic health data of 3 million patients that visited TMU Hospital, Wan Fang Hospital, and Shuang Ho Hospital. We enrolled patients with CKD after coronary stenting between 2008 and 2019. The patients were divided into the long-term (>6 months) and short-term DAPT group (≤ 6 months). The primary end point was major adverse cardiovascular events (MACE) from 6 months after the index date. The secondary outcomes were all-cause mortality and Thrombolysis in Myocardial Infarction (TIMI) bleeding.

**Funding:** The study was supported by the funding and grants from the Taiwan Ministry of Science and Technology (MOST 107-2314-B-038-019-MY3), and Taipei Medical University (109TMU-TMUH-22).

**Competing interests:** The authors have declared that no competing interests exist.

## Results

A total of 1899 patients were enrolled; of them, 1112 and 787 were assigned to the long-term and short-term DAPT groups, respectively. Long-term DAPT was associated with similar risk of MACE (HR: 1.05, 95% CI: 0.65–1.70, $P$ = 0.83) compare with short-term DAPT. Different CKD risk did not modify the risk of MACE. There was also no significant difference in all-cause mortality (HR: 1.10, 95% CI: 0.75–1.61, $P$ = 0.63) and TIMI bleeding (HR 1.19, 95% CI: 0.86–1.63, $P$ = 0.30) between groups.

## Conclusions

Among patients with CKD and coronary stenting, we found that long-term and short-term DAPT tied on the risk of MACE, all-cause mortality and TIMI bleeding.

## Introduction

Dual antiplatelet therapy (DAPT), comprising of aspirin and a P2Y12 inhibitor, lowers the risk of ischemic events in patients with coronary artery disease (CAD) and stent implantation [1, 2]. The efficacy of these combinations to prevent ischemic events was well established, however, the optimal duration of DAPT remained elusive and differs according to the type of coronary event and stent. Both the American Heart Association and European Society of Cardiology guidelines suggest at least 6 months of DAPT after stenting implantation for stable CAD and at least 3 months DAPT for high bleeding risk patients [3–5]. And, some scoring system recommended according to the guideline (e.g. PRECISE-DAPT score) to identify the high bleeding risk patients [5]. However, the use of these scoring systems in chronic kidney disease (CKD) patients was limited because those have been developed based on randomized control trials (RCTs) that excluded advanced CKD patients [6–8]. Furthermore, fewer evidences existed in CKD patients since these patients were under-represented in large clinical trials, though CKD patients are at risk of developing CAD [9].

CKD patients have been reported to have increased risk of ischemia events post stent implantation [10, 11]. The increased risk of ischemia in CKD patients not only comes from old age and comorbidities, but also non-traditional risk factors, namely, chronic inflammation, malnutrition, vascular calcification, and endothelial dysfunction [12, 13]. The increased blood coagulation profiles and increased factor VIII activity were also characterized in CKD patients [14]. In addition, the attenuated response of antiplatelet therapy further increased the risk of ischemic events in these patients [15]. Therefore, prolonged DAPT was considered a logic strategy for CKD patients post-stent implantation. However, those patients are also at higher risk of bleeding events, including gastrointestinal (GI) or cerebrovascular bleeding [16, 17]; and prolonged DAPT may associated with higher risk of bleeding [18]. The bleeding leads to a sudden discontinuation of DAPT, and this adverse event impacts on quality of life and CV-related morbidities [19]. Until now, there has been no consensus to judiciously determine the balance between ischemia and bleeding.

In CKD patients, whether a longer duration of DAPT associated with better CV outcomes, as compared to short DAPT is of interest. Therefore, we aimed to compare the effectiveness and safety between long-term and short-term DAPT in patients with CKD risk after receiving percutaneous coronary intervention (PCI) by a real-world data analysis. Using the Taipei

Medical University (TMU)-Institutional and Clinical Database enables the study to reflect real-world practice involving CKD with different severity.

## Material and methods

### Ethical statement

This study has been reviewed by the Institutional Review Board of Taipei Medical University (TMU-JIRB No. N201707020). The need for informed consent was waived owing to the use of anonymized data.

### Data source

Data were obtained from the Taipei medical University Institutional and Clinical Database [20, 21], which stores electronic health data of 3 million patients that visited TMU Hospital, Wan Fang Hospital, and Shuang Ho Hospital. These hospitals have a combined capacity of 3000 beds. The database contains information regarding patients' demographic and clinical characteristics, outpatient visits, emergency room visits, hospital admissions, laboratory tests results, and drug prescriptions since 1997.

### Patients

We included 10900 patients who underwent PCI between January 1, 2008 and December 31, 2019. The index date was defined as the first date by which PCI with implantation of drug-eluting stents or bare-metal stents was performed. The exclusion criteria were as follows: 1) PCI without stent implantation; 2) missing information in CKD stage or stage G1-G2 with albuminuria $< 30$ mg/g; 3) DAPT not given within 28 days from the index date; 4) $< 28$ days or $> 336$ days DAPT users; 5) death, ischemic stroke or myocardial infarction within 168 days from the index date; 6) receiving warfarin or non-vitamin K antagonist oral anticoagulants (Fig 1). CKD risk on the index date was defined according to the estimated glomerular filtration rate (eGFR) and the amount of microalbuminuria for $> 3$ months, consistent with the Kidney Disease Outcomes Quality Initiative of the National Kidney Foundation definition and classification of CKD [22]. The eGFR was calculated using the serum creatinine (sCr) based on the Taiwanese modification of the Modification of Diet in Renal Disease (MDRD) equation: $1.309 \times MDRD^{0.912}$ [23]. MDRD was calculated as follows: $175 \times sCr^{-1.154} \times age^{-0.203} \times 0.742$ (if female). All diagnoses and procedures from 1997 to 2015 and since 2016 were defined according to the International Classification of Diseases, Ninth Revision, Clinical Modification (ICD-9-CM) and the International Statistical Classification of Diseases and Related Health Problems 10th Revision (ICD-10), respectively (S1 Table). Major, minor and minimal bleeding were defined according to the Thrombolysis In Myocardial Infarction (TIMI) definition [24]. The aim of this study is to compare the effectiveness and safety between long-term and short-term DAPT in patients with CKD risk after coronary stenting.

### Exposure

The patients were divided into two groups according to the duration of DAPT as the long-term DAPT group ($>6$ months) and the short-term DAPT group ($\leq 6$ months). Medications prescribed were retrieved from pharmaceutical data and classified according to the Anatomical Therapeutic Chemical classification system of the World Health Organization (https://www.whocc.no). The use of DAPT was defined as the continuing use of DAPT comprising aspirin plus a platelet $P2Y_{12}$ inhibitor (clopidogrel or ticagrelor) (S1 Table).

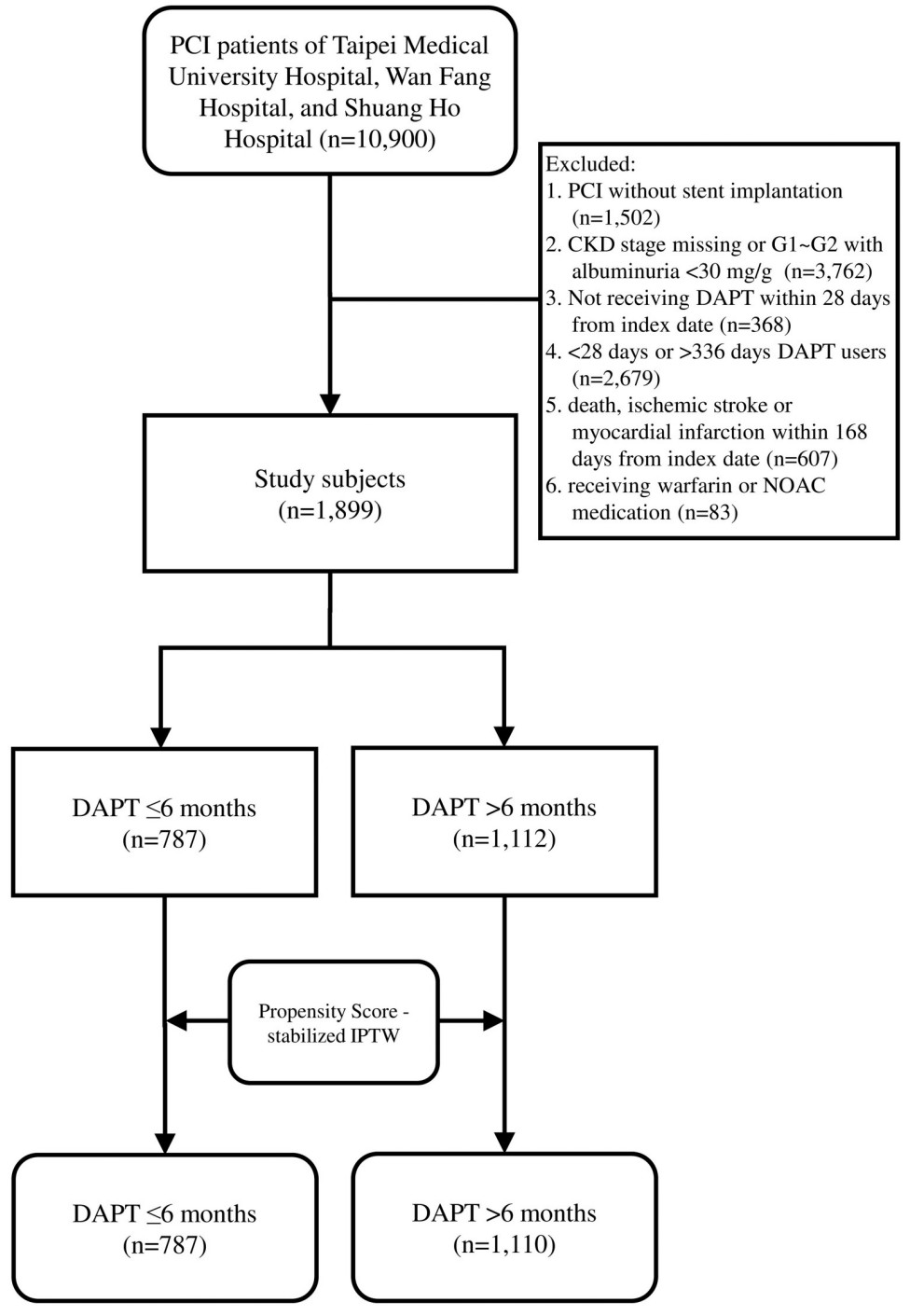

**Fig 1. Flow chart of study subjects selection.**

## Study endpoints

We conducted a landmark analysis, for immortal time bias, to examine the effect of major adverse cardiac event (MACE), all-cause mortality, and bleeding. MACE was defined as a composite of CV death, ischemic stroke, and myocardial infarction. All endpoints were started

to record 6 months after the index date, and the follow-up assessment comprised at least four records of outpatient or inpatient management every year. We linked (TMU) Institutional and Clinical Database with the "Cause of Death Data (death registry)" in Ministry of Health and Welfare by patients' "identification number" [25]. The primary endpoint was the occurrence of MACE. The secondary endpoints were all-cause mortality and TIMI bleeding. TIMI major bleeding included intracranial hemorrhage, a decrease in hemoglobin concentration more than 5 g/dL, or an absolute decrease in hematocrit of at least 15%. TIMI minor bleeding included a decrease in hemoglobin concentration more than 3 g/dL and less than 5 g/dL or an absolute decrease in hematocrit of at least 10% when blood loss was observed and a decrease in hemoglobin concentration of at least 4 g/dL or an absolute decrease in hematocrit of at least 12%. TIMI minimal bleeding included any overt bleeding event that does not meet the criteria above [26]. The follow-up period started from 6 months after the index date until the occurrence of outcomes of interest, 18 months after the index date, or until August 31, 2016, whichever came first.

## Covariates

We collected information regarding sex, age, CKD risk, stent type, coronary events type, and comorbidities (hypertension, diabetes mellitus, hyperlipidemia, peripheral arterial occlusion disease (PAOD), congestive heart failure (CHF), stroke, atrial fibrillation, and cancer) (S1 Table). Baseline comorbidities were identified from two or more records in either outpatient or admission claims within a 2-year period preceding the index date. Complex PCI was defined as chronic total occlusion, bifurcation, triple vessel disease, left main, or ≥3 stents implanted. They were then weighted to calculate the Charlson Comorbidity Index (CCI) excluding comorbidities as mentioned above [27]. In addition, we analyzed drugs used within a 2-year period prior to the index date, including angiotensin-converting enzyme inhibitor (ACEI), angiotensin receptor blocker (ARB), beta-blocker, proton pump inhibitor (PPI), and statins. Laboratory results, namely, eGFR, BUN, albumin, white blood cell, hemoglobin, platelet, sodium, potassium, calcium, phosphorus, parathyroid hormone, glycated hemoglobin, C-reactive protein, and N-terminal-proB-type natriuretic peptide, within a 6-month period prior to the index date were collected. We also analyzed individual patient's PRECISE-DAPT score, which was developed to predict the risk of bleeding in patients with coronary stenting and DAPT [8].

## Statistical analysis

Inverse probability of treatment weighting (IPTW) method, one of the propensity score methods that can be used without reducing power in a full cohort, was used to reduce confounding factors, as in observational studies [28–30]. The marginal probability of receiving exposure was derived from a logistic regression model by using baseline covariates of age, sex, CKD risk, stents type, coronary events type, year of the index date, length of stay, hypertension, diabetes mellitus, hyperlipidemia, PAOD, CHF, stroke, atrial fibrillation, cancer, CCI, PRECISE-DAPT score, ticagrelor, ACEI, ARB, beta-blocker, PPI, statins, and laboratory results. A standardized difference of < 0.1 suggests that baseline covariates were well balanced.

The risk estimates of MACE, all-cause mortality, and TIMI bleeding were summarized as the 95% confidence intervals (CIs) for incidence rate ratios from the Poisson regression model-weighted IPTW and as hazard ratios (HRs) and 95% CIs from the Cox proportional hazard regression model-weighted IPTW. The time to MACE was described using the Kaplan–Meier method-weighted IPTW and compared using the log-rank test.

Stratified analyses of MACE and bleeding were performed for the following covariates: sex, age, CKD risk, complex PCI, stents type, coronary events type, and PRECISE-DAPT score. Scatter plots, boxplots, and histograms were used to describe the distribution of DAPT duration by eGFR and PRECISE-DAPT score.

All data analyses were performed using SAS Enterprise Guide software, version 7.11 (SAS Institute, Cary, NC, USA). A two-tailed $P$ value of $< 0.05$ was considered statistically significant.

## Results

### Baseline patient characteristics

In total, 1899 patients were included in the study; of them, 1112 and 787 patients were categorized to the long-term DAPT group and the short-term DAPT group, respectively. After IPTW, 795 (71.6%) and 563 (71.5%) of patients in the long-term and the short-term DAPT groups were male, respectively, and the mean (SD) age of the patients was 68.6 (12.0) and 68.7 (12.0), respectively. There were 598 (31.5%), 782 (41.2%), 296 (15.6%), and 221 (11.6%) of patients with CKD low, moderate, high and very high risk, respectively. 174 (15.6%) and 127 (16.2%) of patients in the long-term and short-term DAPT groups received stent implantation due to AMI in index admission. Meanwhile, 608 (54.7%) and 434 (55.1%) of patients in the long-term and the short-term DAPT groups had PRECISE-DAPT score $\geq$ 25, respectively. Baseline covariates after IPTW were well-balanced between the two groups. Other baseline characteristics, underlying comorbidities, index admission laboratory data and medication histories are presented in Table 1. In addition, the scattered DAPT duration according to eGFR and PRECISE-DAPT score were drawn in S1 and S2 Figs.

### Association between DAPT duration and risk of MACE and TIMI bleeding

Long-term DAPT was associated with similar risk of MACE (HR: 1.05, 95% CI: 0.65–1.70, $P$ = 0.83) compared with short-term DAPT (Table 2). Further stratification according to myocardial infarction, ischemic stroke and CV death showed comparable risks between long-term and short-term DAPT. No significant differences in all-cause mortality (HR: 1.10, 95% CI: 0.75–1.61, $P$ = 0.63) and risk of TIMI bleeding (HR: 1.19, 95% CI: 0.86–1.63, P = 0.30) between the two groups. There were no significant differences for TIMI bleeding in either TIMI major (HR: 1.63, 95% CI: 0.32–8.35, P = 0.56), TIMI minor (HR: 0.89, 95% CI: 0.43–1.85, P = 0.76) or TIMI minimal bleeding (HR: 1.24, 95% CI: 0.86–1.78, P = 0.25) between long and short DAPT group.

Among the patients with CKD, two groups had comparable risk of MACE (log-rank $P$ = 0.82) (Fig 2A). In our stratified analysis according to the PRECISE-DAPT score and DAPT duration were shown in Fig 2B. Meanwhile, as kidney function worsened, the risk of MACE significantly increased (log-rank $P$ < .0001) (Fig 2C). Among the patients with different CKD risk, long DAPT and short DAPT had no significant difference in MACE (low; HR:2.02, 95% CI: 0.75–5.41, moderate; HR:0.68, 95% CI: 0.31–1.50, high; HR: 1.13, 95% CI: 0.37–3.50, very high; HR: 1.12, 95% CI: 0.44–2.85) (Fig 2D).

The hazard ratio of MACE and bleeding were simultaneously increased with increasing PRECISE-DAPT score (Fig 3). The forest plots for the hazard ratio of MACE and bleeding between groups was shown in Fig 4. In this analysis, no subgroup patients had significant increased risks.

**Table 1. Demographics of patients with percutaneous coronary intervention.**

| | Full Cohort | | | Stabilized IPTW | | |
|---|---|---|---|---|---|---|
| | DAPT ≤6 m (n = 787) | DAPT >6 m (n = 1,112) | ASD* | DAPT ≤6 m (n = 787) | DAPT >6 m (n = 1,110) | ASD* |
| | N (%) | N (%) | | N (%) | N (%) | |
| Sex | | | | | | |
| Male | 530 (67.3) | 829 (74.6) | 0.16 | 563 (71.5) | 795 (71.6) | 0.00 |
| Female | 257 (32.7) | 283 (25.4) | 0.16 | 224 (28.5) | 315 (28.4) | 0.00 |
| Age, year, mean(SD) | 70.1 (11.7) | 67.3 (12.1) | 0.23 | 68.7 (12.0) | 68.6 (12.0) | 0.01 |
| <45 | 18 (2.3) | 48 (4.3) | 0.11 | 25 (3.2) | 38 (3.4) | 0.01 |
| 45–54 | 68 (8.6) | 131 (11.8) | 0.10 | 84 (10.6) | 115 (10.4) | 0.01 |
| 55–64 | 177 (22.5) | 278 (25.0) | 0.06 | 181 (23.0) | 261 (23.5) | 0.01 |
| 65–74 | 228 (29.0) | 343 (30.8) | 0.04 | 241 (30.6) | 337 (30.3) | 0.01 |
| 75–84 | 218 (27.7) | 239 (21.5) | 0.15 | 188 (23.9) | 269 (24.2) | 0.01 |
| ≥85 | 78 (9.9) | 73 (6.6) | 0.12 | 67 (8.6) | 91 (8.2) | 0.01 |
| CKD risk† | | | | | | |
| Low | 247 (31.4) | 365 (32.8) | 0.03 | 247 (31.4) | 351 (31.6) | 0.01 |
| Moderate | 306 (38.9) | 471 (42.4) | 0.07 | 324 (41.2) | 458 (41.2) | 0.00 |
| High | 130 (16.5) | 166 (14.9) | 0.04 | 124 (15.8) | 172 (15.5) | 0.01 |
| Very high | 104 (13.2) | 110 (9.9) | 0.10 | 92 (11.7) | 129 (11.6) | 0.00 |
| Complex PCI‡ | 256 (32.5) | 396 (35.6) | 0.07 | 264 (33.6) | 378 (34.1) | 0.01 |
| Stents type | | | | | | |
| Drug-eluting stents | 400 (50.8) | 728 (65.5) | 0.30 | 473 (60.1) | 663 (59.8) | 0.01 |
| Bare-metal stents | 387 (49.2) | 384 (34.5) | 0.30 | 314 (39.9) | 447 (40.2) | 0.01 |
| Disease type | | | | | | |
| AMI | 76 (9.7) | 216 (19.4) | 0.28 | 127 (16.2) | 174 (15.6) | 0.01 |
| STEMI | 31 (3.9) | 107 (9.6) | 0.23 | 62 (7.9) | 83 (7.5) | 0.02 |
| NSTEMI | 45 (5.7) | 109 (9.8) | 0.15 | 65 (8.2) | 91 (8.2) | 0.00 |
| Non-AMI | 711 (90.3) | 896 (80.6) | 0.28 | 660 (83.8) | 936 (84.4) | 0.01 |
| Previous or coexisting medical condition | | | | | | |
| HTN | 458 (58.2) | 530 (47.7) | 0.21 | 402 (51.1) | 575 (51.8) | 0.01 |
| DM | 275 (34.9) | 361 (32.5) | 0.07 | 256 (32.6) | 364 (32.8) | 0.01 |
| Hyperlipidemia | 342 (43.5) | 394 (35.4) | 0.17 | 297 (37.8) | 421 (38.0) | 0.00 |
| PAOD | 21 (2.7) | 33 (3.0) | 0.02 | 23 (3.0) | 33 (3.0) | 0.00 |
| CHF | 133 (16.9) | 148 (13.3) | 0.10 | 119 (15.1) | 166 (14.9) | 0.01 |
| Stroke | 56 (7.1) | 65 (5.8) | 0.05 | 49 (6.3) | 74 (6.7) | 0.02 |
| Atrial fibrillation | 29 (3.7) | 21 (1.9) | 0.11 | 21 (2.6) | 28 (2.5) | 0.01 |
| Cancer | 39 (5.0) | 35 (3.1) | 0.09 | 30 (3.8) | 42 (3.8) | 0.00 |
| CCI, mean(SD) | 0.85 (1.23) | 0.69 (1.08) | 0.14 | 0.75 (1.16) | 0.76 (1.11) | 0.01 |
| 0 | 417 (53.0) | 687 (61.8) | 0.18 | 457 (58.0) | 640 (57.7) | 0.01 |
| 1–2 | 305 (38.8) | 344 (30.9) | 0.17 | 272 (34.6) | 384 (34.6) | 0.00 |
| ≥3 | 65 (8.3) | 81 (7.3) | 0.04 | 58 (7.4) | 86 (7.7) | 0.01 |
| PRECISE-DAPT score, mean(SD) | 29.2 (12.3) | 27.1 (12.2) | 0.17 | 28.1 (12.2) | 28.1 (12.3) | 0.00 |
| <25 | 318 (40.4) | 547 (49.2) | 0.18 | 353 (44.9) | 502 (45.3) | 0.01 |
| ≥25 | 469 (59.6) | 565 (50.8) | 0.18 | 434 (55.1) | 608 (54.7) | 0.01 |
| Medications | | | | | | |
| Aspirin | 787 (100.0) | 1,112 (100.0) | | 787 (100.0) | 1,110 (100.0) | |
| Clopidogrel | 757 (96.2) | 991 (89.1) | 0.27 | 725 (92.2) | 1,020 (91.9) | 0.01 |
| Ticagrelor | 95 (12.1) | 307 (27.6) | 0.40 | 167 (21.2) | 235 (21.2) | 0.00 |
| ACEI | 81 (10.3) | 91 (8.2) | 0.07 | 69 (8.8) | 101 (9.1) | 0.01 |

*(Continued)*

**Table 1.** (Continued)

| | Full Cohort | | | Stabilized IPTW | | |
|---|---|---|---|---|---|---|
| | DAPT ≤6 m (n = 787) | DAPT >6 m (n = 1,112) | ASD* | DAPT ≤6 m (n = 787) | DAPT >6 m (n = 1,110) | ASD* |
| | N (%) | N (%) | | N (%) | N (%) | |
| ARB | 439 (55.8) | 476 (42.8) | 0.26 | 382 (48.5) | 537 (48.4) | 0.00 |
| Beta-blocker | 467 (59.3) | 506 (45.5) | 0.29 | 405 (51.5) | 566 (51.0) | 0.01 |
| PPI | 121 (15.4) | 155 (13.9) | 0.04 | 111 (14.2) | 162 (14.6) | 0.01 |
| Statins | 396 (50.3) | 480 (43.2) | 0.14 | 359 (45.6) | 507 (45.7) | 0.00 |
| Laboratory | | | | | | |
| eGFR, g/dL, ml/min per 1.73 m², mean(SD) | 46.6 (24.2) | 48.9 (20.9) | 0.10 | 47.4 (23.5) | 47.5 (21.4) | 0.01 |
| Hb, g/dL, mean(SD) | 13.4 (1.7) | 13.7 (1.6) | 0.19 | 13.5 (1.6) | 13.7 (1.7) | 0.02 |
| WBC, x10³/uL, mean(SD) | 8.7 (3.9) | 9.2 (3.6) | 0.19 | 8.8 (4.0) | 9.2 (3.6) | 0.00 |
| Albumin, g/dL, mean(SD) | 3.9 (0.3) | 3.9 (0.3) | 0.05 | 3.9 (0.3) | 3.9 (0.3) | 0.04 |
| Na, mmol/L, mean(SD) | 139.9 (3.7) | 139.4 (3.0) | 0.14 | 139.9 (3.6) | 139.5 (2.9) | 0.10 |
| K, mmol/L, mean(SD) | 4.4 (0.6) | 4.3 (0.6) | 0.18 | 4.4 (0.6) | 4.3 (0.6) | 0.07 |
| Ca, mg/dL, mean(SD) | 9.0 (0.4) | 9.0 (0.4) | 0.03 | 9.0 (0.4) | 9.0 (0.4) | 0.05 |
| P, mg/dL, mean(SD) | 4.6 (0.8) | 4.5 (0.7) | 0.07 | 4.6 (0.7) | 4.6 (0.8) | 0.01 |
| PTH, pg/mL, mean(SD) | 231 (35) | 230 (32) | 0.03 | 230 (31) | 231 (32) | 0.03 |
| HbA₁c, %, mean(SD) | 6.9 (1.1) | 6.8 (1.2) | 0.06 | 6.9 (1.0) | 6.9 (1.2) | 0.00 |
| NT-ProBNP, pg/mL, mean(SD) | 1,124 (1,128) | 1,025 (952) | 0.10 | 1,102 (1,108) | 1,047 (973) | 0.05 |
| CRP, mg/dL, mean(SD) | 1.5 (1.7) | 1.5 (2.3) | 0.02 | 1.5 (1.7) | 1.5 (2.3) | 0.02 |
| BUN, mg/dL, mean(SD) | 31.9 (27.0) | 29.1 (24.8) | 0.11 | 31.0 (25.4) | 30.3 (26.3) | 0.03 |
| PLT, x10³/uL, mean(SD) | 227.1 (66.9) | 228.5 (70.9) | 0.02 | 228.2 (66.8) | 227.4 (70.5) | 0.01 |

*Absolute standardized difference >0.1 for imbalance.

†Low: eGFR ≥60 with albuminuria 30–300 or eGFR 45–59 with albuminuria <30; moderate: eGFR ≥60 with albuminuria ≥300 or eGFR 45–59 with albuminuria 30–300 or eGFR 30–44 with albuminuria <30; high: eGFR 45–59 with albuminuria ≥300 or eGFR 30–44 with albuminuria ≥30 or eGFR <30; very high: dialysis.

‡Complex PCI: chronic total occlusion, bifurcation, triple vessel disease, left main, or ≥3 stents implanted.

Abbreviation: IPTW, Inverse Probability of Treatment Weighting; ASD, absolute standardized difference; DAPT, dual antiplatelet therapy; CKD, chronic kidney disease; DES, Drug-eluting stents; BMS, Bare-metal stents; PCI, percutaneous coronary intervention; AMI, acute myocardial infarction; STEMI, ST-elevation myocardial infarction; NSTEMI, non-ST-elevation myocardial infarction; HTN, Hypertension; DM, Diabetes mellitus; PAOD, peripheral arterial occlusion disease; CHF, congestive heart failure; CCI, Charlson comorbidity index; ACEI, angiotensin converting enzyme inhibitor; ARB, angiotensin receptor blocker; PPI, proton pump inhibitor; Hb, hemoglobin; Na, sodium; K, potassium; Ca, calcium; P, phosphorus; HbA1c glycated hemoglobin; NT-ProBNP, pro B-Type Natriuretic Peptide.

## Discussion

This study compared the effectiveness and safety between long-term (> 6 months) and short-term (≤6 months) DAPT after coronary stenting in patients with CKD. The following key findings were observed: (1) After IPTW adjustment, there was no significant difference in MACE outcome between long-term and short-term DAPT group and the same for the risk of TIMI bleeding (2) With the progress of CKD, increased MACE event was noted in patients with more advanced CKD. Furthermore, the risk of MACE and bleeding simultaneously increased as the PRECISE-DAPT score increased. (3) There was no statistically significant difference in MACE and TIMI bleeding rate between long and short term DAPT group in subgroup analysis including age, CKD risk, index admission disease type, stent type and PRECISE-DAPT score.

CKD is considered in the context of increased ischemic and bleeding risk that makes the decision of DAPT duration difficult. There are no reliable scoring systems to predict the risk of future events [31]. Optimal duration of DAPT after coronary artery stenting in CKD

**Table 2. Outcomes of PCI patients between DAPT >6 months and ≤6 months after IPTW.**

| | N | Event | Person months | IR per 1,000 person-months (95% CI) | IR ratio (95% CI) | HR (95% CI) | p |
|---|---|---|---|---|---|---|---|
| **Primary outcome** | | | | | | | |
| MACE | 1,897 | 88 | 30,112 | 2.9 (2.3–3.6) | | | |
| DAPT ≤6 m | 787 | 34 | 12,012 | 2.8 (1.9–3.9) | 1.00 (ref.) | 1.00 (ref.) | |
| DAPT >6 m | 1,110 | 54 | 18,100 | 3.0 (2.2–3.9) | 1.06 (0.69–1.63) | 1.05 (0.65–1.70) | 0.83 |
| CV death | 1,897 | 32 | 30,511 | 1.0 (0.7–1.5) | | | |
| DAPT ≤6 m | 787 | 13 | 12,146 | 1.1 (0.6–1.9) | 1.00 (ref.) | 1.00 (ref.) | |
| DAPT >6 m | 1,110 | 19 | 18,365 | 1.0 (0.6–1.6) | 0.91 (0.45–1.85) | 0.91 (0.43–1.90) | 0.80 |
| Ischemic stroke | 1,897 | 41 | 30,317 | 1.4 (1.0–1.8) | | | |
| DAPT ≤6 m | 787 | 14 | 12,097 | 1.2 (0.6–2.0) | 1.00 (ref.) | 1.00 (ref.) | |
| DAPT >6 m | 1,110 | 27 | 18,220 | 1.5 (1.0–2.2) | 1.26 (0.66–2.39) | 1.25 (0.63–2.48) | 0.52 |
| Myocardial infarction | 1,897 | 16 | 30,483 | 0.5 (0.3–0.8) | | | |
| DAPT ≤6 m | 787 | 6 | 12,139 | 0.5 (0.2–1.1) | 1.00 (ref.) | 1.00 (ref.) | |
| DAPT >6 m | 1,110 | 9 | 18,344 | 0.5 (0.2–1.0) | 1.02 (0.37–2.82) | 1.02 (0.29–3.63) | 0.98 |
| **Secondary outcome** | | | | | | | |
| All-cause mortality | 1,897 | 121 | 30,291 | 4.0 (3.3–4.8) | | | |
| DAPT ≤6 m | 787 | 45 | 12,114 | 3.7 (2.7–5.0) | 1.00 (ref.) | 1.00 (ref.) | |
| DAPT >6 m | 1,110 | 75 | 18,178 | 4.1 (3.3–5.2) | 1.11 (0.77–1.60) | 1.10 (0.75–1.61) | 0.63 |
| TIMI bleeding | 1,897 | 186 | 29,234 | 6.4 (5.5–7.4) | | | |
| DAPT ≤6 m | 787 | 67 | 11,694 | 5.7 (4.4–7.3) | 1.00 (ref.) | 1.00 (ref.) | |
| DAPT >6 m | 1,110 | 119 | 17,539 | 6.8 (5.6–8.1) | 1.19 (0.88–1.60) | 1.19 (0.86–1.63) | 0.30 |
| TIMI major | 1,897 | 9 | 30,549 | 0.3 (0.1–0.5) | | | |
| DAPT ≤6 m | 787 | 3 | 12,170 | 0.2 (0.0–0.7) | 1.00 (ref.) | 1.00 (ref.) | |
| DAPT >6 m | 1,110 | 6 | 18,380 | 0.3 (0.1–0.7) | 1.66 (0.38–7.17) | 1.63 (0.32–8.35) | 0.56 |
| TIMI minor | 1,897 | 32 | 30,385 | 1.0 (0.7–1.5) | | | |
| DAPT ≤6 m | 787 | 13 | 12,102 | 1.1 (0.6–1.9) | 1.00 (ref.) | 1.00 (ref.) | |
| DAPT >6 m | 1,110 | 18 | 18,282 | 1.0 (0.6–1.6) | 0.90 (0.44–1.81) | 0.89 (0.43–1.85) | 0.76 |
| TIMI minimal | 1,897 | 146 | 29,496 | 4.9 (4.2–5.8) | | | |
| DAPT ≤6 m | 787 | 51 | 11,792 | 4.3 (3.2–5.7) | 1.00 (ref.) | 1.00 (ref.) | |
| DAPT >6 m | 1,110 | 95 | 17,704 | 5.4 (4.3–6.5) | 1.24 (0.88–1.74) | 1.24 (0.86–1.78) | 0.25 |

*MACE: ischemic stroke, myocardial infarction, or CV death.

Abbrevation: CI, confidence interval; CV, cardiovascular; DAPT, dual antiplatelet therapy; HR, hazard ratio; IR, incidence rate; MACE, major adverse cardiac event; NA, not applicable; PCI, percutaneous coronary intervention; TIMI, thrombolysis in myocardial infarction.

patients remains unclear as large randomized control trials were limited of evidences. There were only some post-hoc analysis and retrospective studies focusing on CKD populations. Two post-hoc analysis studies have reported short-term DAPT use may reduce the risk of bleeding without increasing ischemic event in CKD patients [18, 32]. On the contrary, a national cohort in Taiwan targeting dialysis patients and a meta-analysis enrolling patients with moderate CKD both found short-term DAPT use was not associated with lower bleeding rate or higher MACE event [33, 34]. In our study, we found both MACE and bleeding outcome were not significantly different between long-term and short-term DAPT group. There are some main reasons to explain the difference between our study and previous studies. Prior studies only focused on dialysis and drug-eluting stent patients. Our study includes diverse stage of CKD patients, including low CKD risk patients with eGFR>60 and moderately increased albuminuria. Different types of stents were also included. There were few studies to

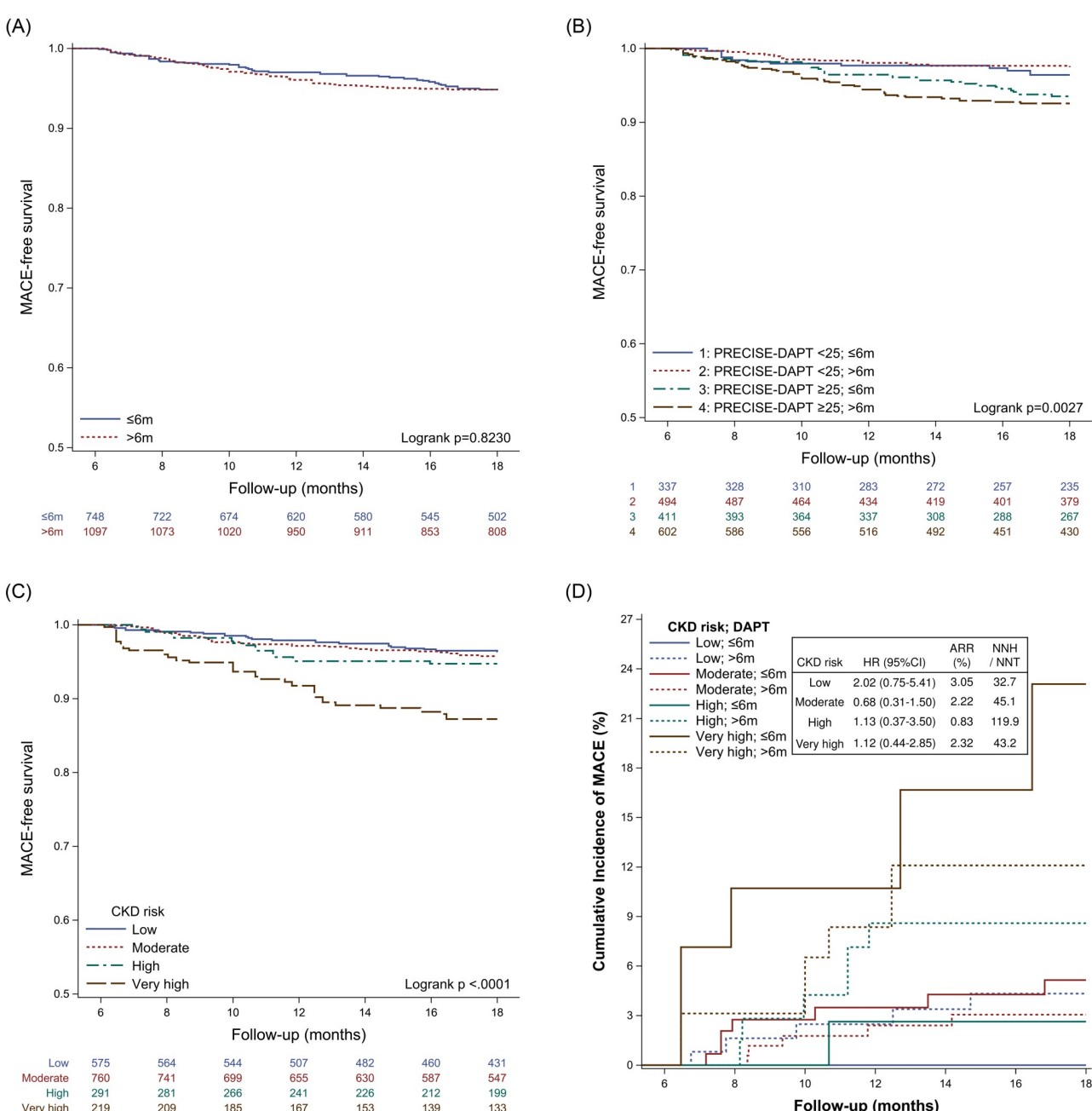

**Fig 2. Kaplan-Meier plot of MACE and cumulative incidence of MACE after IPTW: (A) DAPT duration; (B) PRECISE-DAPT score and DAPT duration; (C) CKD risk; (D) CKD risk and DAPT duration.**

discuss the low-risk CKD population, especially considering the microalbuminuria level. Understanding those low CKD risk population's optimal DAPT duration is crucial because the prior evidences showed increased risk of developing CAD in the patients with eGFR below 75 ml/min/1.73 m2 or with microalbuminuria [35, 36].

In subgroup analysis, we discovered that there was no significant difference regarding MACE and bleeding events in different ages or CKD risk, drug-eluting stent vs. bare-metal

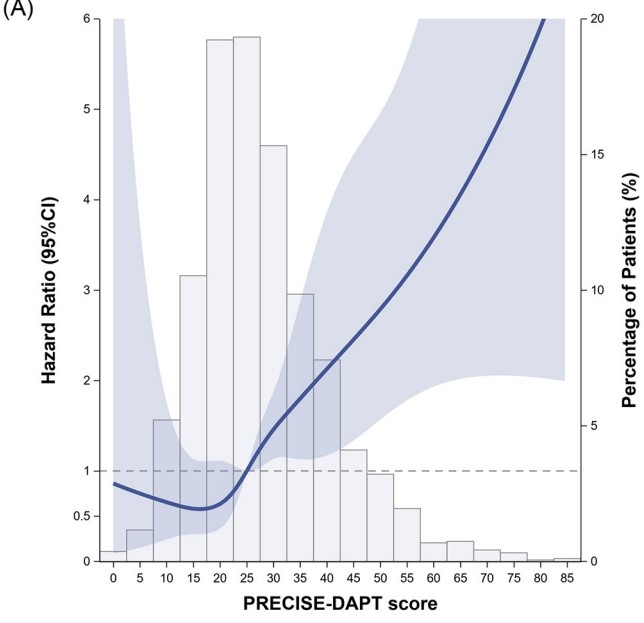

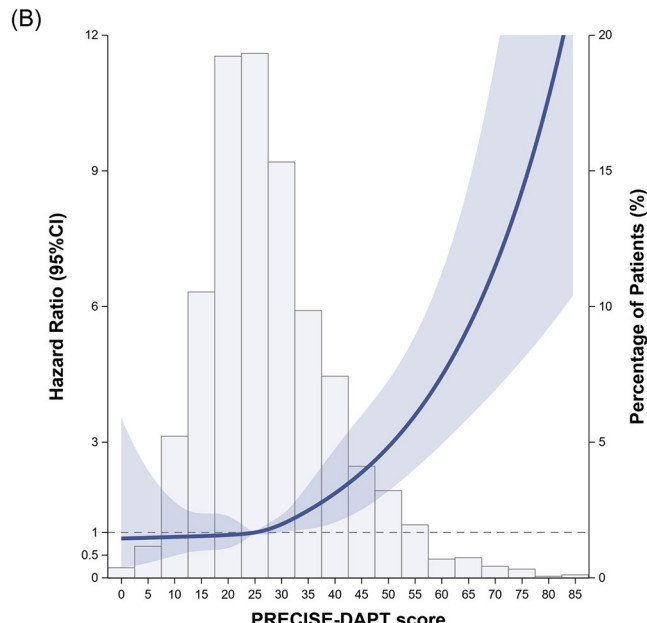

**Fig 3. The associations between PRECISE-DAPT score and the hazard for the MACE and bleeding using restricted cubic splines after IPTW with five knots at the 5th, 27.5th, 50th, 72.5th, and 95th percentiles: (A) MACE; (B) bleeding.**

stent, AMI vs. Non-AMI. Recently, several studies that included advanced CKD or AMI patients have advocated the efficacy and safety of short DAPT use 3–6 months [1, 37, 38] or 1 month [39, 40]. Collectively, those findings along with these in the current study indicate that short DAPT use might be an alternative option.

High platelet reactivity and poor response to oral antiplatelet therapies may explain why long-term DAPT fails to be associated with lower risk of MACE events and short-term DAPT

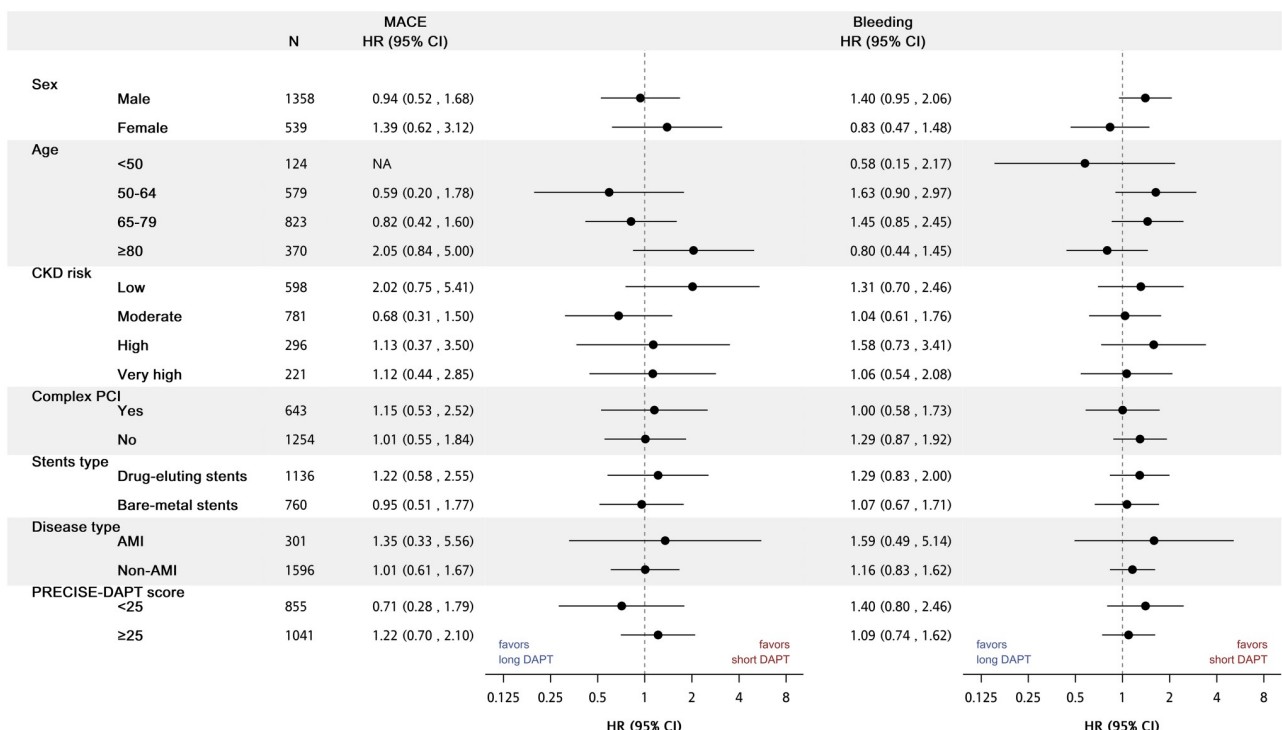

**Fig 4. Forest plot depicting the risk of MACE and bleeding associated with DAPT >6 months and ≤6 months after IPTW.** Abbreviation: AMI, acute myocardial infarction; Cl, confidence interval; CKD, chronic kidney disease; HR, hazard ratios; MACE, major adverse cardiac event; PCI, percutaneous coronary intervention. *p<0.05, **p<0.001.

not related to lower bleeding episode in CKD patients. Previous studies showed that with respect to platelet inhibition, clopidogrel has inferior efficacy to prasugrel or ticagrelor in CKD patients [15, 41, 42]. In other word, in clopidogrel-based DAPT, CKD patients may not obtain the benefit of desirable CV outcomes compared to patients with normal renal function [43, 44]. Another possible reason which DAPT fails to reduce MACE event significantly could be poor metabolizer of CYP2C19 which has higher incidence in Asian population especially in Taiwanese [45]. In total, 95% of the patients in the current study used clopidogrel-based DAPT regimen, and this could partly explain why long-term DAPT did not reduce MACE events.

Another merit of the current study is to find that the PRECISE-DAPT score might not be an ideal tool to decide the duration of DAPT in CKD patients. Unlike prior PRECISE-DAPT score analysis showing longer DAPT exerted an ischemia benefit in lower PRECISE-DAPT score group (<25) in general population [8]; our study showed CKD patients with PRECISE-DAPT score < 25 had no significant difference of MACE outcome between long and short DAPT use. Moreover, there was no significant difference in bleeding for higher PRECISE-DAPT score group (≥25) between long and short DAPT, either. Our study found that both the hazard ratio of bleeding and MACE increased as the PRECISE-DAPT score increased; however, it is noted that PRECISE-DAPT score was only effective for bleeding risk and MACE prediction in CKD patients but not an effective predictive tool for DAPT duration determination.

There are several limitations in our study. First, the retrospective nature of the study limited the understanding for individual physician's decision regarding DAPT duration. The

discovery in this study might not represent the whole CKD population. However, it should still be substantial for the investigation regarding to DAPT use in CKD patients as the population were usually excluded in prior RCTs, and the current study could be one more piece in that puzzle via real world data analysis. Second, 95% of our patients were under clopidogrel treatment, however, some evidence has shown that new P2Y12 inhibitors, like ticagrelor or prasugrel, are more effective in preventing MACE events in CKD patients compared to clopidogrel [46]. A different result might be possible if more CKD patients using ticagrelor or prasugrel were included in the study for analysis. Third, the bleeding events might have been underestimated because of the use of a retrospective population-based cohort and an institutional clinical database; patients may have transferred to a different institution or did not comply with the planned follow-up visits after PCI. To eliminate the impact of this limitation, we assessed the outcomes only in patients who were followed up for at least four times within 1 year to ensure the follow up.

## Conclusions

In this population-based cohort study, we found that among CKD patients undergoing coronary stenting, long-term and short-term DAPT tied on the risk of MACE and all-cause mortality. There was no significant difference in TIMI bleeding outcome between long-term and short-term DAPT patients.

## Supporting information

**S1 Fig. Scatter plots, boxplots, and histograms of DAPT duration by eGFR.**
(TIF)

**S2 Fig. Scatter plots, boxplots, and histograms of DAPT duration by PRECISE-DAPT score.**
(TIF)

**S1 Table. Diagnosis codes and medication codes.**
(DOCX)

## Acknowledgments

We thank the staff of the Office of Information Technology, Taipei Medical University, for technical support.

## Author Contributions

**Conceptualization:** Mai-Szu Wu, Chih-Wei Chen, Tzu-Hao Chang.

**Data curation:** Ming-Tsang Chuang, Tzu-Hao Chang.

**Supervision:** Mai-Szu Wu, Yi-Cheng Lin, Chun-Yao Huang, Wei-Chiao Chang, Tzu-Hao Chang.

**Writing – original draft:** Chih-Chin Kao.

**Writing – review & editing:** Chih-Wei Chen.

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
