## [Decision Letter · Decision Letter 0]

17 Jun 2021

PONE-D-21-13103

Investigation of dual antiplatelet therapy after coronary stenting in patients with chronic kidney diseas e

PLOS ONE

Dear Dr. chen,

Thank you for submitting your manuscript to PLOS ONE. After careful consideration, we feel that it has merit but does not fully meet PLOS ONE’s publication criteria as it currently stands. Therefore, we invite you to submit a revised version of the manuscript that addresses the points raised during the review process.

We look forward to receiving your revised manuscript.

Kind regards,

Hitesh Raheja, MD

Academic Editor

PLOS ONE

Journal Requirements:

2. Thank you for including your methods section:  "This study has been reviewed by the Institutional Review Board of TMU (TMU-JIRB No. N201707020). The need for informed consent was waived owing to the use of anonymized data."   

1. Please add this information to your ethics statement in the online submission form.

2. Please amend your current ethics statement to confirm that your named institutional review board or ethics committee specifically approved this study.

3. Thank you for providing the date(s) when patient medical information was initially recorded. Please also include the date(s) on which your research team accessed the databases/records to obtain the retrospective data used in your study.

Reviewers' comments:

Reviewer's Responses to Questions

**Comments to the Author**

1. Is the manuscript technically sound, and do the data support the conclusions?

Reviewer #1: Yes

Reviewer #2: Yes

2. Has the statistical analysis been performed appropriately and rigorously? 

Reviewer #1: Yes

Reviewer #2: Yes

3. Have the authors made all data underlying the findings in their manuscript fully available?

Reviewer #1: Yes

Reviewer #2: Yes

4. Is the manuscript presented in an intelligible fashion and written in standard English?

Reviewer #1: Yes

Reviewer #2: Yes

5. Review Comments to the Author

Reviewer #1: Methods: Confusing ? as this is a retrospective chart review and the authors have mentioned they enrolled the patients, please make it clearer what is study type and data collected?

Fix both in abstract and methods

Define the clear aim in manuscript in methods

Reviewer #2: This is an interesting article about DAPT use in CKD patients. The authors demonstrated that longer (>6 months) vs. shorter (< 6 months) have similar MACE and TIMI bleeding risk. This article is in line with the recent published data about the safety of shorter DAPT. CKD patients are prone to bleeding but also have higher rates of ACS and ischemic events, which makes this paper interesting for readers.

6. PLOS authors have the option to publish the peer review history of their article (what does this mean?). If published, this will include your full peer review and any attached files.

Reviewer #1: **Yes: **Shyam Odeti

Reviewer #2: No

---

## [Author Response · Author response to Decision Letter 0]

30 Jun 2021

Journal Requirements

Comment 1

Response: Thank you for the kind reminder. We checked the references and confirmed no retracted paper in the reference list.

Comment 2

Response: We follow your suggestion to revise the manuscript to meet PLOS ONE’s style requirements. We change level 1 heading to 18pt font and level 2 heading to 16pt font. And we also change our figure and table citation format to meet the style of PLOS ONE. 

Comment 3

Thank you for including your methods section: "This study has been reviewed by the Institutional Review Board of TMU (TMU-JIRB No. N201707020). The need for informed consent was waived owing to the use of anonymized data." 

1. Please add this information to your ethics statement in the online submission form.

2. Please amend your current ethics statement to include the full name of the ethics committee/institutional review board(s) that approved your specific study. Once you have amended this/these statement(s) in the Methods section of the manuscript, please add the same text to the “Ethics Statement” field of the submission form (via “Edit Submission”).

Response: Thank you for the comment. We have added this ethics statement in the online submission form and we also revised our description in method section (page 7, line 2, method section) to “This study has been reviewed by the Institutional Review Board of Taipei Medical University (TMU-JIRB No. N201707020). The need for informed consent was waived owing to the use of anonymized data.”

Comment 4

Thank you for providing the date(s) when patient medical information was initially recorded. Please also include the date(s) on which your research team accessed the databases/records to obtain the retrospective data used in your study.

Response: Thank you for the comment. Our research team applied to access Taipei Medical University Institutional and Clinical Database after this study was approved by the Institutional Review Board of Taipei Medical University. Our first analysis of this database was on February 06, 2020.

Reviewers’ comments

Reviewer #1: 

Methods: Confusing ? as this is a retrospective chart review and the authors have mentioned they enrolled the patients, please make it clearer what is study type and data collected? Fix both in abstract and methods. Define the clear aim in manuscript in methods. 

Response: Thank you for the valuable comment. Our study is a retrospective cohort study use anonymized data from Taipei Medical University (TMU) Institutional and Clinical Database. This database included 3 million patients’ electronic medical record form TMU Hospital, Wan Fang Hospital, and Shuang Ho Hospital respectively. The aim of our study is to compare the effectiveness and safety between long-term and short-term dual anti-platelet therapy in patients with chronic kidney disease after coronary stenting. 

 We revised the description in abstract (page 3, line 9, abstract section) to “This retrospective cohort study analyze data from the Taipei Medical University (TMU) Institutional and Clinical Database, which include anonymized electronic health data of 3 million patients that visited TMU Hospital, Wan Fang Hospital, and Shuang Ho Hospital. We also added the aim of our study in methods section (page 8, line 13) “The aim of this study is to compare the effectiveness and safety between long-term and short-term DAPT in patients with CKD risk after coronary stenting.”

Reviewer #2: 

This is an interesting article about DAPT use in CKD patients. The authors demonstrated that longer (>6 months) vs. shorter (< 6 months) have similar MACE and TIMI bleeding risk. This article is in line with the recent published data about the safety of shorter DAPT. CKD patients are prone to bleeding but also have higher rates of ACS and ischemic events, which makes this paper interesting for readers

Response: Thank you for the comment. CKD patients are considered to have higher ischemia and bleeding risk compare to general populations. This is the reason why we want to conduct this study to compare the effectiveness and safety between long-term and short-term DAPT in these patients in real world setting.

---

## [Editor Report · Decision Letter 1]

21 Jul 2021

Investigation of dual antiplatelet therapy after coronary stenting in patients with chronic kidney diseas e

PONE-D-21-13103R1

Dear Dr. chen,

We’re pleased to inform you that your manuscript has been judged scientifically suitable for publication and will be formally accepted for publication once it meets all outstanding technical requirements.

Kind regards,

Hitesh Raheja, MD

Academic Editor

PLOS ONE
---

## [Editor Report · Acceptance letter]

26 Jul 2021

PONE-D-21-13103R1 

Investigation of dual antiplatelet therapy after coronary stenting in patients with chronic kidney disease 

Dear Dr. Chen:

I'm pleased to inform you that your manuscript has been deemed suitable for publication in PLOS ONE. Congratulations! Your manuscript is now with our production department. 

Kind regards, 

on behalf of

Dr. Hitesh Raheja 

Academic Editor

PLOS ONE